*Experimental Results* (2021), 2, e4, 1–9

# Simplifying the Measurement of College Students' Career Planning: the Development of Career Student Planning Scale during the COVID-19 Pandemic

Magnus A. Gray[1], Minsung Kim[2] and Seungyeon Lee[1,*]

[1]School of Social & Behavioral Sciences, University of Arkansas at Monticello, 562 University Drive, Monticello, Arkansas 71656, USA, and [2]Defense Language Institute Foreign Language Center, Presidio of Monterey, Monterey, CA 93944, USA
*Corresponding author. E-mail: lees@uamont.edu

(Received 14 November 2020; Revised 13 December 2020; Accepted 13 December 2020)

## Abstract

We created a new, 8-item scale called "Career Student Planning Scale (CSPS)" for a valid and reliable measure regarding college students' career planning during a traumatic event, such as a pandemic. CSPS is conceptually similar to the career decision-making difficulty questionnaire (CDDQ) and the career decision self-efficacy (CDSE) scale. CSPS leans towards questions about college students' perceptions about career planning, rather than intuitions about career decision-making; it also inquires about how participants conceptualize about their career plans to be correct, rather than the more extreme idea about how their intuitions are correct: we developed this scale to capture the latter construct. We included the coronavirus anxiety scale (CAS), CDDQ, the general procrastination scale (GPS), and the CDSE short form (CDSE-SF) as covariates to ensure that CSPS has distinct effects on their career paths. Our findings indicate the CSPS has acceptable psychometric properties and demonstrates a valuable input to those measures.

**Keywords:** COVID-19 anxiety; career decision self-efficacy (CDSE); career student planning scale (CSPS); procrastination

## 1. Introduction

Given the extensive threats caused by the COVID-19 pandemic, college students' mental health and career planning have become critical priorities across university campuses. Due to the novelty of this pandemic, little has been investigated on the relationship between coronavirus-related anxiety and young adults' career decision making. Mahmud et al. (2020) found that fear of COVID-19 impacts career decisions through its inherent depression issues. This coincides with other empirical findings that show (a) young adults are more anxious about their careers, and (b) this anxiety has an impact on future career plans (Campagna & Curtis, 2007; Mojgan et al., 2011). Furthermore, Işik's (2012) study showed there is a significantly negative relationship between career decision-making self-efficacy and trait anxiety. Together, these studies indicate that anxiety affects both young adults and college students with their careers, which is why we examine whether coronavirus anxiety affects both career planning and indecision. Career measures, like the career decision-making difficulty questionnaire (CDDQ) and the career decision-making self-efficacy scale (short form) (CDSE-SF) are widely used psychometric assessments, but how researchers quantify young adults' attitudes regarding career planning has scarcely been explored (Gati et al., 1996; Gray et al., 2020; Lee et al., 2018; Tomaszek & Muchacka-Cymerman, 2020).

## 2. Objective

Psychometric assessment has an important role in health assessment and research. We created the 8-item college student planning scale (CSPS) to better quantify these variables. CSPS examines college students' career plans to identify those affected with the pandemic's uncertainty with most issues, including occupations and careers. We also investigated the degree to which CSPS relates to pre-existing and validated measures (i.e., the coronavirus anxiety scale [CAS], CDDQ, general procrastination scale [GPS], and the career decision self-efficacy scale [CDSE-SF]) during the recent pandemic. We examined the CSPS's factorial structure and internal validity to better establish psychometric validation of the measure.

## 3. Methods

Participants ($N$ =101) were originally recruited at small, liberal arts colleges in southeast Arkansas and southern California, except that one failed to answer some questions—which was excluded. Each participant voluntarily participated in the study, which was approved by the Institutional Review Board (IRB) at the University of Arkansas at Monticello and carried out with APA ethical standards. All participants received a copy of the consent form at the beginning of the study, and we provided debriefing after its completion. They completed the CSPS, CDDQ, GPS, and CDSE-SF in an online format and received extra credit for their involvement. The order of each questionnaire was randomized per participant to avoid an order effect. One-hundred participants were included for data analysis ($M$ = 22.25, $SD$ = 6.6; 77 females and 22 males). There were 25 Black (or African-American), 67 White, and 7 identifying with "others." Four were listed as college freshmen, 34 were sophomores, 26 were juniors, and 33 were seniors enrolled full-time. Two listed having post-baccalaureate degrees.

A simple regression analysis was conducted to see how CSPS relates to CDDQ and CDSE-SF. The coefficient alpha of CSPS was 0.85, which is a desirable value as a measure, according to Cortina (1993). The CSPS' significant correlation coefficients used solid planning ($r$ = .29, p < .01) and management ($r$ = .28, p < .01) showing potential validity evidence of its content, whether it is valid to the extent of planning management. After assessing reliability and validity of the CSPS, a regression analysis was conducted for the impact of CAS on CSPS. Two simple regressions of the CSPS were conducted to see the impact of the CDDQ and CDSE-SF.

## 4. Results

In all analyses, data from the CSPS, CAS, CDDQ, GPS, and CDSE-SF were used as continuous variables. Correlation coefficients among variables are shown in Table 1. CAS consists of 5 items, so we assume the measure has a tendency to assess anxiety related to COVID-19. Results of the first simple linear regression indicates that the effect between the CAS and CSPS was insignificant. The skewness value, 3.24, indicates a positively skewed response trend. Significant findings were assessed when examining the CSPS's correlations with CDDQ, GPS, and CDSE (see Table 1).

**Table 1.** Correlations between CSPS, CDSE-SF, GPS, and CAS

|  | CSPS | CDSE-SF | GPS | CDDQ |
|---|---|---|---|---|
| CDSE-SF | 0.52*** |  |  |  |
| GPS | -0.27** | -0.29** |  |  |
| CDDQ | -0.47*** | -0.33*** | 0.12 |  |
| CAS | -0.07 | -0.06 | 0.04 | 0.16 |

p < .001 '***', p < .01 '**', p < .05 '*'

**Table 2.** Internal consistency estimates of CSPS, CDSE-SF, GPS, CDDQ, and CAS

|  | No. of items | Coefficient alpha | Item scale |
|---|---|---|---|
| CSPS | 8 | 0.85 | 1-5 |
| CDSE-SF | 25 | 0.95 | 1-5 |
| GPS | 20 | 0.79 | 1-5 |
| CDDQ | 32 | 0.93 | 1-9 |
| CAS | 5 | 0.96 | 0-4 |

We also examined the factor structure among the given measures. Table 2 shows the item-level descriptive scale and internal consistency estimates among CSPS, CDSE-SF, GPS, CDDQ, and CAS. In the two regression analyses for CSPS, the new measure had a strong impact on CDDQ and CDSE-SF. The first CSPS simple linear regression indicated a significant effect on CDSE-SF, ($F_{(1, 98)} = 37.19$, $p < .001$, $R^2 = .28$) and how CSPS was a strong predictor ($t = 6.1$, $p < .001$) of the model. The second CSPS-SF simple linear regression concluded there was a similarly significant effect on CDDQ, ($F_{(1, 98)} = 27.67$, $p < .001$, $R^2 = .22$) and that CSPS was also a significant predictor ($t = -5.3$, $p < .001$) of the model.

## 5. Discussion

The current study examined the correlation between CSPS, CAS, CDDQ, GPS, and CDSE-SF. We first examined how COVID-19 anxiety affects college students and career planning; however, no correlation was shown between them. The results suggest that CAS is case-sensitive, so we caution against its use as a measure of anxiety for the general population. The CAS was newly developed in April 2020 when COVID-19 peaked and participants were a bit older (Lee, 2020). Previous literature shows a relationship between anxiety and career indecision (Campagna & Curtis, 2007; Işik, 2012; Mojgan et al., 2011; Tomaszek & Muchacka-Cymerman, 2020), so future research must focus on the correlation between generalized anxiety and career planning. More independent studies that investigate the psychometric properties of the CSPS are also necessary to establish the scientific rigor of this research.

No significant relationship was found between the CAS and CSPS, but our findings suggest moderate-sized correlations between CSPS-SF and CDDQ, and GPS and CDSE. A significant negative correlation was found between CSPS and CDDQ ($r = -.47$, $p < .001$). This suggests that college students will face fewer career difficulties if they plan for their future careers, possibly with some backup vocations. We found that the CSPS is negatively correlated with the GPS ($r = -.27$, $p < .001$), suggesting that factors related to procrastination may instigate career planning as well as decision difficulties. A significant positive relationship was found between the CSPS and CDSE-SF ($r = .52$, $p < .001$), indicating how adequate planning for a future career may allow college students to feel more confident about this decision. Overall, our study shows that CSPS can be a reliable, unidimensional construct for career planning in university samples, along with the general population.

## 6. Conclusion and Future Directions

This study serves as a psychometric analysis of the CSPS during the COVID-19 pandemic. Coronavirus-related anxiety has been a serious issue affecting young adults' career planning and decisions, so we attempted to assess if COVID-19 anxiety affects the relationship between CSPS, CDDQ, GPS, and CDSE-SF among U.S. college students. The CAS was created for researchers and health professionals to identify anxiety and uncertainty in this growing pandemic, so the quality of our measures will hopefully provide solid evidence of how these measures were in fact examined, helping researchers select the best assessment plan.

One possible explanation about not finding a significant relationship between CAS and CSPS could be that young adults do not have an excessive fear of death, on which the 5 items of CAS specifically focus. All items ask about one's physiologically-based anxiety reaction to COVID-19-related symptoms, which many young adults may not experience. Future studies will examine generalized anxiety with these measures about whether generalized anxiety is predictive of the actual population, rather than their current involvement with the recent pandemic.

We found interesting relationships between the CSPS and other measures in this study (i.e., the CDDQ, GPS, and CDSE-SF). In conclusion, the CSPS is a promising measure with relevance to college students. The higher scores of the CDSE correspond to higher levels of CSPS, which may contribute to growing mental symptoms, suggesting that without proper planning, this can lead to distraction in daily situations, such as college learning.

The current study represents a measurement of career planning, which will help us conduct future research related to college students' mental health and professional development, with a mediation-based personality theory and career framework. The assessment of psychometric properties promotes the selection of valid and reliable instruments, so that researchers can ensure the internal validity of their results. An empirical investigation on how COVID-19-related anxiety impacts college students and their career plans should contribute to the current literature and to future researchers involved with the COVID-19 pandemic.

**Acknowledgements.** We thank Drs. Jennifer Miller and Hyowon Ban for assistance with data collection, as well as Dr. Carol Zurcher for carefully reviewing and editing the manuscript. We are deeply grateful to them for their encouragement and support.

**Conflict of interest.** The authors declare no conflicts of interest.

**Funding information.** This work was supported by the Arkansas Department of Higher Education as a form of student undergraduate research fellowship [SURF] (MG & SL, grant number 544060-2100-212111).

**Author contributions.** MG, MK, and SL conceived and designed the study. MG and SL conducted data collection, MG and MK performed statistical analysis, and MG, MK, and SL wrote the article.

**Data availability statement.** Data for the experiments that is reported here is available upon request. The experiment was not preregistered.

**Compliance with ethical standards.** The authors assert that all procedures contributing to this work comply with the ethical standards of the relevant national and institutional committees on human experimentation and with the Helsinki Declaration of 1975, revised in 2008.

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

# Peer Reviews

## Reviewing editor: Dr. Xiaoping Wang

Second Xiangya Hospital, Department of psychiatry, 139 Renmin Middle Road, Changsha, Hunan, China, 410011

This article has been accepted because it is deemed to be scientifically sound, has the correct controls, has appropriate methodology and is statistically valid, and has been sent for additional statistical evaluation and met required revisions.

doi:10.1017/exp.2020.69.pr1

## Review 1: Simplifying the Measurement of College Students' Career Planning: the Development of Career Student Planning Scale during the COVID-19 Pandemic

### Reviewer: Dr. Teresa Ober

University of Notre Dame, Department of Psychology, E418 Corbett Family Hall, Notre Dame, Indiana, United States, 46556

Date of review: 18 November 2020

**Conflict of interest statement.** Reviewer declares none.

*Comments to the Author:* Thank you for focusing your research question on such a timely and applied problem. I could see several clear uses and implications of these findings. Nevertheless, I have a few hesitations about the current draft as outlined below.

Introduction

-Given the brevity of the article, I think the introduction as written is fairly successful in justifying the study and providing background.

Methods:

-Further details about the participants would be helpful (e.g., age, gender, class they were enrolled in or college majors, etc.). This could be included as a table to spare the word count.

-Were the questions randomized or asked in a fixed order? I ask because it could impact an order effect.

-Since part of the aim of this paper is to examine the factor structure, it would be great to see information about the item-level scale descriptives and internal consistency estimates.

Conclusion

-I am not sure what this sentence means and was hoping for a further explanation, perhaps with a citation: "No significant relationship between CAS and our newly developed measure (i.e., the CSPS) was found due to the time sensitivity of CAS.

Minor:

-Some grammatical issues here and there. For example,

--"CSPS leans toward questions..." I think should read, "CSPS leans towards questions…"

--"CAS developed by Lee (2020) is also relative to a new measure,…" I think should read, "CAS developed by Lee (2020) is also a relaively new measure…"

**Score Card**

Presentation

3.6
/5

| | |
|---|---|
| Is the article written in clear and proper English? (30%) | 4/5 |
| Is the data presented in the most useful manner? (40%) | 3/5 |
| Does the paper cite relevant and related articles appropriately? (30%) | 4/5 |

Context

4.2
/5

| | |
|---|---|
| Does the title suitably represent the article? (25%) | 5/5 |
| Does the abstract correctly embody the content of the article? (25%) | 4/5 |
| Does the introduction give appropriate context? (25%) | 4/5 |
| Is the objective of the experiment clearly defined? (25%) | 4/5 |

Analysis

3.4
/5

| | |
|---|---|
| Does the discussion adequately interpret the results presented? (40%) | 3/5 |
| Is the conclusion consistent with the results and discussion? (40%) | 4/5 |
| Are the limitations of the experiment as well as the contributions of the experiment clearly outlined? (20%) | 3/5 |

doi:10.1017/exp.2020.69.pr2

# Review 2: Simplifying the Measurement of College Students' Career Planning: the Development of Career Student Planning Scale during the COVID-19 Pandemic

**Reviewer:** Dr. Jinyao Yi 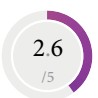

Date of review: 07 December 2020

**Conflict of interest statement.** Reviewer declares none

*Comments to the Author:* 1）The references cited in the introducyion seem susceptible, for example, how could the two early studies in 2007 and 2011 address COVID-19-related depression/anxiety and their impact on students' future career planning? The introduction could be benefited greatly through involving more studies that were conducted in recent years and have used more reliable research methods to explain how anxiety impacts career decision-making, instead of simply leaving one corre-lational conclusion obtained in 2012. If such evidence is unavailable, then the importance of the whole topic will be in question.

2) The introduction does not specify on how the new scale is different from other scales (CDDQ, CDSE-SF) and why the authors bother to develop a new scale. What unique advantages can CSPS bring, at least theoretically? What academic or practical gap does this 8-item scale fill?

3) The introduction and objective sections are repetitive. Please integrate the contents into one section.

4) The authors just recruited 100 participants. Is this sample size predefined or naturally obtained? If it's predefined, please justify it (i.e. the method of calculating sample size).

5) What method on earth did the authors use to demonstrate the unifactorial structure of CSPS?

6) Please provide detailed descriptions for all the scales used in this manuscript. It is also necessary to report the internal consistency reliability of each measurement found in the current study.

7) How can the correlations with these two factors justify the validity of the new scale? Please provide more robust evidence to establish the validity and move all the calculations in methods to the results section.

8) Due to the disproportionate gender composition, it is necessary to control gender as a covariance in the regression analysis.

9) Please provide ethical details for this study.

10) I would hesitate calling the CSPS "a psychometrically robust measure… in college samples from general population" based on the current evidence, since the psychometric properties of this scale remain questionable based on the current sample size and statistical analysis.

## Score Card
### Presentation

| | | |
|---|---|---|
| **2.6**/5 | Is the article written in clear and proper English? (30%) | 3/5 |
| | Is the data presented in the most useful manner? (40%) | 2/5 |
| | Does the paper cite relevant and related articles appropriately? (30%) | 3/5 |

## Context

3.0
/5

| | |
|---|---|
| Does the title suitably represent the article? (25%) | 4/5 |
| Does the abstract correctly embody the content of the article? (25%) | 2/5 |
| Does the introduction give appropriate context? (25%) | 3/5 |
| Is the objective of the experiment clearly defined? (25%) | 3/5 |

## Analysis

2.8
/5

| | |
|---|---|
| Does the discussion adequately interpret the results presented? (40%) | 3/5 |
| Is the conclusion consistent with the results and discussion? (40%) | 3/5 |
| Are the limitations of the experiment as well as the contributions of the experiment clearly outlined? (20%) | 2/5 |