## [Reviewer Report · Review 1: Simplifying the Measurement of College Students’ Career Planning: the Development of Career Student Planning Scale during the COVID-19 Pandemic]

*Comments to the Author:* Thank you for focusing your research question on such a timely and applied problem. I could see several clear uses and implications of these findings. Nevertheless, I have a few hesitations about the current draft as outlined below.

Introduction

-Given the brevity of the article, I think the introduction as written is fairly successful in justifying the study and providing background.

Methods:

-Further details about the participants would be helpful (e.g., age, gender, class they were enrolled in or college majors, etc.). This could be included as a table to spare the word count.

-Were the questions randomized or asked in a fixed order? I ask because it could impact an order effect.

-Since part of the aim of this paper is to examine the factor structure, it would be great to see information about the item-level scale descriptives and internal consistency estimates.

Conclusion

-I am not sure what this sentence means and was hoping for a further explanation, perhaps with a citation: “No significant relationship between CAS and our newly developed measure (i.e., the CSPS) was found due to the time sensitivity of CAS.

Minor:

-Some grammatical issues here and there. For example,

--“CSPS leans toward questions...” I think should read, “CSPS leans towards questions…”

--“CAS developed by Lee (2020) is also relative to a new measure,…” I think should read, “CAS developed by Lee (2020) is also a relaively new measure…”

## Score Card

### Presentation

3.6/5

Is the article written in clear and proper English?30%4/5

Is the data presented in the most useful manner?40%3/5

Does the paper cite relevant and related articles appropriately?30%4/5

### Context

4.2/5

Does the title suitably represent the article?25%5/5

Does the abstract correctly embody the content of the article?25%4/5

Does the introduction give appropriate context?25%4/5

Is the objective of the experiment clearly defined?25%4/5

### Analysis

3.4/5

Does the discussion adequately interpret the results presented?40%3/5

Is the conclusion consistent with the results and discussion?40%4/5

Are the limitations of the experiment as well as the contributions of the experiment clearly outlined?20%3/5

---

## [Reviewer Report · Review 2: Simplifying the Measurement of College Students’ Career Planning: the Development of Career Student Planning Scale during the COVID-19 Pandemic]

*Comments to the Author:* 1）The references cited in the introducyion seem susceptible, for example, how could the two early studies in 2007 and 2011 address COVID-19-related depression/anxiety and their impact on students’ future career planning? The introduction could be benefited greatly through involving more studies that were conducted in recent years and have used more reliable research methods to explain how anxiety impacts career decision-making, instead of simply leaving one correlational conclusion obtained in 2012. If such evidence is unavailable, then the importance of the whole topic will be in question.

2) The introduction does not specify on how the new scale is different from other scales (CDDQ, CDSE-SF) and why the authors bother to develop a new scale. What unique advantages can CSPS bring, at least theoretically? What academic or practical gap does this 8-item scale fill?

3) The introduction and objective sections are repetitive. Please integrate the contents into one section.

4) The authors just recruited 100 participants. Is this sample size predefined or naturally obtained? If it’s predefined, please justify it (i.e. the method of calculating sample size).

5) What method on earth did the authors use to demonstrate the unifactorial structure of CSPS?

6) Please provide detailed descriptions for all the scales used in this manuscript. It is also necessary to report the internal consistency reliability of each measurement found in the current study.

7) How can the correlations with these two factors justify the validity of the new scale? Please provide more robust evidence to establish the validity and move all the calculations in methods to the results section.

8) Due to the disproportionate gender composition, it is necessary to control gender as a covariance in the regression analysis.

9) Please provide ethical details for this study.

10) I would hesitate calling the CSPS “a psychometrically robust measure… in college samples from general population” based on the current evidence, since the psychometric properties of this scale remain questionable based on the current sample size and statistical analysis.

## Score Card

### Presentation

2.6/5

Is the article written in clear and proper English?30%3/5

Is the data presented in the most useful manner?40%2/5

Does the paper cite relevant and related articles appropriately?30%3/5

### Context

3.0/5

Does the title suitably represent the article?25%4/5

Does the abstract correctly embody the content of the article?25%2/5

Does the introduction give appropriate context?25%3/5

Is the objective of the experiment clearly defined?25%3/5

### Analysis

2.8/5

Does the discussion adequately interpret the results presented?40%3/5

Is the conclusion consistent with the results and discussion?40%3/5

Are the limitations of the experiment as well as the contributions of the experiment clearly outlined?20%2/5